# Exact Representation of Sparse Networks with Symmetric Nonnegative Embeddings

**Sudhanshu Chanpuriya**[1]**, Ryan A. Rossi**[2]**, Anup Rao**[2]**, Tung Mai**[2]**,
Nedim Lipka**[2]**, Zhao Song**[2]**, and Cameron Musco**[3]

[1]University of Illinois Urbana-Champaign, `schariya@illinois.edu`
[2]Adobe Research, `{ryrossi,anuprao,tumai,lipka,zsong}@adobe.com`
[3]University of Massachusetts Amherst, `cmusco@cs.umass.edu`

## Abstract

Graph models based on factorization of the adjacency matrix often fail to capture network structures related to links between dissimilar nodes (heterophily). We introduce a novel graph factorization model that leverages two nonnegative vectors per node to interpretably account for links between both similar and dissimilar nodes. We prove that our model can exactly represent any graph with low *arboricity*, a property that many real-world networks satisfy; our proof also applies to related models but has much greater scope than the closest prior bound, which is based on low *max degree*. Our factorization also has compelling properties besides expressiveness: due to its symmetric structure and nonnegativity, fitting the model inherently finds node communities, and the model's link predictions can be interpreted in terms of these communities. In experiments on real-world networks, we demonstrate our factorization's effectiveness on a variety of tasks, including community detection and link prediction.

## 1 Introduction

Graphs data naturally arises in a variety of fields including sociology (Mason & Verwoerd, 2007), biology (Scott, 1988), and computer networking (Bonato, 2004). A key task in machine learning for graph data is forming models of graphs that can predict edges between nodes, form useful representations of nodes, and reveal interpretable structure in the graph, such as detecting clusters of nodes. Many graph models fall under the framework of edge-independent graph generative models, which output the probabilities of edges existing between any pair of nodes. The parameters of such models can be trained iteratively on the network, or some fraction of the network which is known, in the link prediction task, i.e., by minimizing a predictive loss. To choose among these models, one must consider two criteria: 1) whether the model can express structures of interest in the graph, 2) whether the model expresses these structures in an interpretable way.

**Expressiveness of low-dimensional embeddings** As real-world graphs are high-dimensional objects, graph models generally compress information about the graph. For example, dot product models associate each node with a real-valued "embedding" vector; the predicted probability of a link between two nodes increases with the similarity of their embeddings. These models can alternatively be seen as factorizing the graph's adjacency matrix to approximate it with a low-rank matrix. Recent work of Seshadri et al. (2020) has shown that dot product models are limited in their ability to model common structures in real-world graphs, such as triangles incident only on low-degree nodes. In response, Chanpuriya et al. (2020) showed that with the logistic principal components analysis (LPCA) model, which has two embeddings per node (i.e., using the dot product of the 'left' embedding of one node and the 'right' embedding of another), not only can such structures be represented, but further, any graph can be exactly represented with embedding vectors whose lengths are linear in the

max degree of the graph. There are two keys to this result. First is the presence of a nonlinear linking function in LPCA; since adjacency matrices are generally not low-rank, exact low-rank factorization is impossible without a linking function. Second is that having two embeddings rather than one allows for expression of non-positive semidefinite (PSD) matrices. As discussed in Peysakhovich & Bottou (2021), that the single-embedding models can only represent PSD matrices precludes representation of 'heterophilous' structures in graphs; heterophilous structures are those wherein dissimilar nodes are linked, in contrast to more intuitive 'homophilous' linking between similar nodes.

**Interpretability and node clustering**    Beyond being able to capture a given network accurately, it is often desirable for a graph model to form interpretable representations of nodes and to produce edge probabilities in an interpretable fashion. Dot product models can achieve this by restricting the node embeddings to be nonnegative. Nonnegative factorization has long been used to decompose data into parts (Donoho & Stodden, 2003). In the context of graphs, this entails decomposing the set of nodes of the network into clusters or communities. In particular, each entry of the nonnegative embedding vector of a node represents the intensity with which the node participates in a community. This allows the edge probabilities output by dot product models to be interpretable in terms of coparticipation in communities. Depending on the model, these vectors may have restrictions such as a sum-to-one requirement, meaning the node is assigned a categorical distribution over communities. The least restrictive and most expressive case is that of soft assignments to overlapping communities, where the entries can vary totally independently. In such models, which include the BIGCLAM model of Yang & Leskovec (2013), the output of the dot product may be mapped through a nonlinear link function (as in LPCA) to produce a probability for each edge, i.e., to ensure the values lie in $[0, 1]$.

**Summary of contributions**    The key contributions of this work are as follows:

- We prove that LPCA admits exact low-rank factorizations of graphs with bounded *arboricity*, which is the minimum number of forests into which a graph's edges can be partitioned. By the Nash-Williams theorem, arboricity is a measure of a graph's density in that, letting $S$ denote an induced subgraph and $n_S$ and $m_S$ denote the number of nodes and edges in $S$, arboricity is the maximum over all $S$ of $\lceil \frac{m_S}{n_S - 1} \rceil$. Our result is more applicable to real-world graphs than the prior result of Chanpuriya et al. (2020) for graphs with bounded max degree, since sparsity is a common feature of real networks, whereas low max degree is not.

- We introduce a graph model that extends LPCA and is both highly expressive and interpretable. Our model incorporates two embeddings per node and a nonlinear linking function, and hence is able to express both heterophily and overlapping communities. At the same time, our model is based on symmetric nonnegative matrix factorization, so it outputs link probabilities that are interpretable in terms of the communities it detects. While prior graph factorizations incorporate some aspects of nonnegativity, heterophily, and/or nonlinearity, our proposed model lies at the intersection of all three.

- We show how any graph with a low-rank factorization in the LPCA model also admits a low-rank factorization in our community-based model. This means that the guarantees on low-rank representation for bounded max degree and arboricity also apply to our model.

- In experiments, we show that our method is competitive with and often outperforms other comparable models on real-world graphs in terms of representing the network, doing interpretable link prediction, and detecting communities that align with ground-truth.

## 2    Motivating Example

To demonstrate how heterophily can manifest in networks, as well as how models which assume homophily can fail to represent such networks, we provide a simple synthetic example involving a graph of matches between users of a dating app. Suppose that users of this app are generally seeking partners of a different gender (to simplify this example, we assume that each user of the app is either a man or a woman), and suppose that each user comes from one of ten cities. Members from the same city are likely to match with each other; this typifies homophily, wherein links occur between similar nodes. Furthermore, users having the same gender are unlikely to match with each other; this typifies heterophily. Figure 1 shows an instantiation of such an adjacency matrix with 1000 nodes, which are randomly assigned to man or woman and to one of the ten cities. We recreate this network with our proposed embedding model and with BIGCLAM, which explicitly assumes

homophily. We also compare with the SVD of the adjacency matrix, which outputs the best (lowest Frobenius error) low-rank approximation that is possible without a nonlinear linking function. Since SVD lacks nonnegativity constraints on the factors, we do not expect intepretability. In Figure 2, we show how BIGCLAM captures only the ten communities based on city, i.e., only the homophilous structure, and fails to capture the heterophilous distinction between men and women. We also plot the error of the reconstructions as the embedding length increases. There are $10 \cdot 2 = 20$ different kinds of nodes, meaning the expected adjacency matrix is rank-20, and our model maintains the lowest error up to this embedding length; by contrast, BIGCLAM is unable to decrease error after capturing city information with length-10 embeddings. In Figure 3, we visualize the features generated by the three methods, i.e., the factors returned by each factorization. Our model's factors capture the relevant latent structure in an interpretable way. By contrast, SVD's factors are harder to interpret, and BIGCLAM does not represent the heterophilous structure.

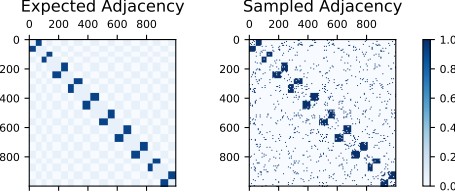

Figure 1: The motivating synthetic graph. The expected adjacency matrix (left) and the sampled matrix (right); the latter, which is passed to the training algorithms, is produced by treating the entries of the former as parameters of Bernoulli distributions and sampling. The network is approximately a union of ten bipartite graphs, each of which correspond to men and women in one of the ten cities.

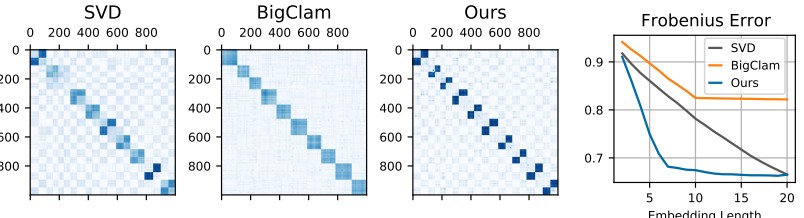

Figure 2: Left: Reconstructions of the motivating synthetic graph of Figure 1 with SVD, BIGCLAM, and our model, using 12 communities or singular vectors. Note the lack of the small diagonal structure in BIGCLAM's reconstruction; this corresponds to its inability to capture the heterophilous interaction between men and women. Right: Frobenius error when reconstructing the motivating synthetic graph of Figure 1 with SVD, BIGCLAM, and our model, as the embedding length is varied. The error is normalized by the sum of the true adjacency matrix (i.e., twice the number of edges).

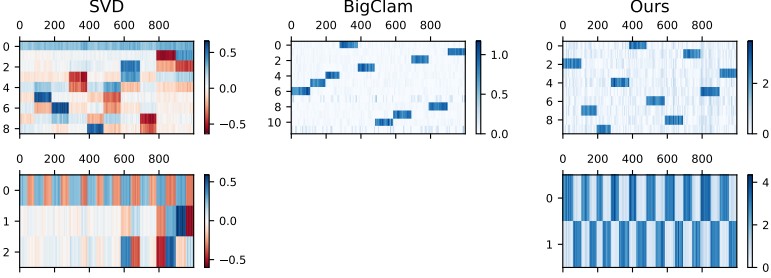

Figure 3: Factors resulting from decomposition of the motivating synthetic graph of Figure 1 with the three models, using 12 communities or singular vectors. The top/bottom rows represent the positive/negative eigenvalues corresponding to homophilous/heterophilous communities (note that BIGCLAM does not include the latter). The homophilous factors from BIGCLAM and our model reflect the 10 cities, and the heterophilous factor from our model reflect men and women. The factors from SVD are harder to interpret. Note that the order of the communities in the factors is arbitrary.

# 3 Community-Based Graph Factorization

Consider the set of undirected, unweighted graphs on $n$ nodes, i.e., the set of graphs with symmetric adjacency matrices in $\{0, 1\}^{n \times n}$. We propose an edge-independent generative model for such graphs. Given nonnegative parameter matrices $\boldsymbol{B} \in \mathbb{R}_+^{n \times k_B}$ and $\boldsymbol{C} \in \mathbb{R}_+^{n \times k_C}$, we set the probability of an edge existing between nodes $i$ and $j$ to be the $(i, j)$-th entry of matrix $\tilde{\boldsymbol{A}}$:

$$\tilde{\boldsymbol{A}} := \sigma(\boldsymbol{B}\boldsymbol{B}^\top - \boldsymbol{C}\boldsymbol{C}^\top), \tag{1}$$

where $\sigma$ is the logistic function. Here $k_B$, $k_C$ are the number of homophilous/heterophilous clusters. Intuitively, if $\boldsymbol{b}_i \in \mathbb{R}_+^{k_B}$ is the $i$-th row of matrix $\boldsymbol{B}$, then $\boldsymbol{b}_i$ is the affinity of node $i$ to each of the $k_B$ homophilous communities. Similarly, $\boldsymbol{c}_i \in \mathbb{R}_+^{k_C}$ is the affinity of node $i$ to the $k_C$ heterophilous communities. As an equivalent statement, for each pair of nodes $i$ and $j$, $\tilde{\boldsymbol{A}}_{i,j} := \sigma(\boldsymbol{b}_i \boldsymbol{b}_j - \boldsymbol{c}_i \boldsymbol{c}_j^\top)$. We will soon discuss the precise interpretation of this model, but the idea is roughly similar to the attract-repel framework of Peysakhovich & Bottou (2021). When nodes $i$ and $j$ have similar 'attractive' $\boldsymbol{b}$ embeddings, i.e., when $\boldsymbol{b}_i \boldsymbol{b}_j^\top$ is high, the likelihood of an edge between them increases, hence why the $\boldsymbol{B}$ factor is homophilous. By contrast, the $\boldsymbol{C}$ factor is 'repulsive'/heterophilous since, when $\boldsymbol{c}_i \boldsymbol{c}_j^\top$ is high, the likelihood of an edge between $i$ and $j$ decreases.

**Alternate expression**    We note that the model above can also be expressed in a form that normalizes cluster assignments and is more compact, in that it combines the homophilous and heterophilous cluster assignments. Instead of $\boldsymbol{B}$ and $\boldsymbol{C}$, this form uses a matrix $\boldsymbol{V} \in [0, 1]^{n \times k}$ and a diagonal matrix $\boldsymbol{W} \in \mathbb{R}^{k \times k}$, where $k = k_B + k_C$ is the total number of clusters. In particular, let $\boldsymbol{m}_B$ and $\boldsymbol{m}_C$ be the vectors containing the maximums of each column of $\boldsymbol{B}$ and $\boldsymbol{C}$. By setting

$$\begin{aligned} \boldsymbol{V} &= \left( \boldsymbol{B} \cdot \operatorname{diag}\left(\boldsymbol{m}_B^{-1}\right); \quad \boldsymbol{C} \cdot \operatorname{diag}\left(\boldsymbol{m}_C^{-1}\right) \right) \\ \boldsymbol{W} &= \operatorname{diag}\left( \left(+\boldsymbol{m}_B^2; \quad -\boldsymbol{m}_C^2\right) \right), \end{aligned} \tag{2}$$

the constraint on $\boldsymbol{V}$ is satisfied. Further, $\boldsymbol{V}\boldsymbol{W}\boldsymbol{V}^\top = \boldsymbol{B}\boldsymbol{B}^\top - \boldsymbol{C}\boldsymbol{C}^\top$, so

$$\tilde{\boldsymbol{A}} := \sigma(\boldsymbol{B}\boldsymbol{B}^\top - \boldsymbol{C}\boldsymbol{C}^\top) = \sigma(\boldsymbol{V}\boldsymbol{W}\boldsymbol{V}^\top). \tag{3}$$

Here, if $\mathbf{v}_i \in [0, 1]^k$ is the $i$-th row of matrix $\boldsymbol{V}$, then $\mathbf{v}_i$ is the soft (normalized) assignment of node $i$ to the $k$ communities. The diagonal entries of $\boldsymbol{W}$ represent the strength of the homophily (if positive) or heterophily (if negative) of the communities. For each entry, $\tilde{\boldsymbol{A}}_{i,j} = \sigma(\mathbf{v}_i \boldsymbol{W} \mathbf{v}_j^\top)$. We use these two forms interchangeably throughout this work.

**Interpretation**    The edge probabilities output by this model have an intuitive interpretation. Recall that there are bijections between probability $p \in [0, 1]$, odds $o = \frac{p}{1-p} \in [0, \infty)$, and logit $\ell = \log(o) \in (-\infty, +\infty)$. The logit of the link probability between nodes $i$ and $j$ is $\mathbf{v}_i^\top \boldsymbol{W} \mathbf{v}_j$, which is a summation of terms $\mathbf{v}_{ic} \mathbf{v}_{jc} \boldsymbol{W}_{cc}$ over all communities $c \in [k]$. If the nodes both fully participate in community $c$, that is, $\mathbf{v}_{ic} = \mathbf{v}_{jc} = 1$, then the edge logit is changed by $\boldsymbol{W}_{cc}$ starting from a baseline of 0, or equivalently, the odds of an edge is multiplied by $\exp(\boldsymbol{W}_{cc})$ starting from a baseline odds of 1; if either of the nodes participates only partially in community $c$, then the change in logit and odds is accordingly prorated. Homophily and heterophily also have a clear interpretation in this model: homophilous communities, which are expressed in $\boldsymbol{B}$, are those with $\boldsymbol{W}_{cc} > 0$, where two nodes both participating in the community increases the odds of a link, whereas communities with $\boldsymbol{W}_{cc} < 0$, which are expressed in $\boldsymbol{C}$, are heterophilous, and coparticipation decreases the odds of a link.

# 4 Related Work

**Community detection via interpretable factorizations**    There is extensive prior work on community detection and node clustering (Schaeffer, 2007; Aggarwal & Wang, 2010; Nascimento & De Carvalho, 2011), perhaps the most well-known being the normalized cuts algorithm of Shi & Malik (2000), which produces a clustering based on the entrywise signs of an eigenvector of the graph Laplacian matrix. However, the clustering algorithms which are most relevant to our work are those based on non-negative matrix factorization (NMF) (Lee & Seung, 1999; Berry et al., 2007; Wang & Zhang, 2012; Gillis, 2020), many of which can be seen as integrating nonnegativity constraints into

the broader, well-studied *random dot product graph* (RDPG) model (Young & Scheinerman, 2007; Scheinerman & Tucker, 2010; Athreya et al., 2017). One such algorithm is that of Yu et al. (2005), which approximately factors a graph's adjacency matrix $A \in \{0,1\}^{n \times n}$ into two positive matrices $H$ and $\Lambda$, where $H \in \mathbb{R}_+^{n \times k}$ is left-stochastic and $\Lambda \in \mathbb{R}_+^{k \times k}$ is diagonal, such that $H \Lambda H^\top \approx A$. Here $H$ represents a soft clustering of the $n$ nodes into $k$ clusters, while the diagonal entries of $\Lambda$ represent the prevalence of edges within clusters. Note the similarity of the factorization to our model, save for the lack of a nonlinearity. Other NMF approaches include those of Ding et al. (2008), Yang et al. (2012), Kuang et al. (2012), and Kuang et al. (2015) (SYMNMF).

**Modeling heterophily**  Much of the existing work on graph models has an underlying assumption of network homophily (Johnson et al., 2010; Noldus & Van Mieghem, 2015). There has been significant recent interest in the limitations of graph neural network (GNN) models (Duvenaud et al., 2015; Kipf & Welling, 2017; Hamilton et al., 2017) at addressing network heterophily (NT & Maehara, 2019; Zhu et al., 2020; Zheng et al., 2022), as well as proposed solutions (Pei et al., 2020; Yan et al., 2021), but relatively less work for models applicable to clustering. Some existing NMF approaches to clustering do naturally model heterophilous structure in networks. The model of Nourbakhsh et al. (2014), for example, is similar to that of Yu et al. (2005), but allows the cluster affinity matrix $\Lambda$ to be non-diagonal; this allows for inter-cluster edge affinity to exceed intra-cluster edge affinity, so heterophily can arise in this model, though it is not a focus of their work. Another example is the model in Miller et al. (2009), which is similar to ours, though it restricts the cluster assignment matrix $V$ to be binary; additionally, their training algorithm is not based on gradient descent as ours is, and it does not scale to larger networks. More recently, Rubin-Delanchy et al. (2017) and Peysakhovich & Bottou (2021) propose simple decompositions which allow for representation of non-PSD adjacency matrices. The model in the latter work is a factorization of the form $A \approx D + BB^\top - CC^\top$, where $D \in \mathbb{R}^{n \times n}$ is diagonal and $B, C \in \mathbb{R}^{n \times k}$ are low-rank; excluding the diagonal $D$ term, the model in the former work is algebraically identical. The authors discuss how, interestingly, this model separates the homophilous and heterophilous structure into different factors, namely $B$ and $C$, corresponding to positive and negative eigenvalues, respectively. Note that these decompositions do not include a nonlinear linking function, which is crucial to our exact factorization results, and the respective works do not investigate constraining the factors to be nonnegative.

**Overlapping communities and exact embeddings**  Many models discussed above focus on the single-label clustering task and thus involve highly-constrained factorizations (e.g., sum-to-one conditions). We are interested in the closely related but distinct task of multi-label clustering, also known as overlapping community detection (Xie et al., 2013; Javed et al., 2018), which involves less constrained, more expressive factorizations. The BIGCLAM algorithm of Yang & Leskovec (2013) uses the following generative model for this task: the probability of a link between two nodes $i$ and $j$ is given by $1 - \exp(-\boldsymbol{f}_i \cdot \boldsymbol{f}_j)$, where $\boldsymbol{f}_i, \boldsymbol{f}_j \in \mathbb{R}_+^k$ represent the intensities with which the nodes participate in each of the $k$ communities. Note that BIGCLAM assumes strict homophily of the communities: two nodes participating in the same community always increases the probability of a link. However, this model allows for expression of very dense intersections of communities, which the authors observe is generally a characteristic of real-world networks. To ensure that output entries are probabilities, BIGCLAM's factorization includes a nonlinear linking function (namely, $f(x) = 1 - e^x$), like our model and LPCA. Recent work outside clustering and community detection on graph generative models (Rendsburg et al., 2020; Chanpuriya et al., 2020) suggests that incorporating a nonlinear linking function can greatly increase the expressiveness of factorization-based graph models, to the point of being able to exactly represent a graph. This adds to a growing body of literature on expressiveness guarantees for embeddings on relational data (Sala et al., 2018; Bhattacharjee & Dasgupta, 2020; Boratko et al., 2021). Most relevant to our work, as previously discussed, Chanpuriya et al. (2020) provide a guarantee for exact low-rank representation of graphs with bounded max degree when using the LPCA factorization model. In this work, we provide a new such guarantee, except for bounded arboricity, which is more applicable to real-world networks, and extend these guarantees to our community-based factorization.

## 5   Theoretical Results

We first restate the main result from Chanpuriya et al. (2020) on exact representation of graphs with bounded max degree using the logistic principal components analysis (LPCA) model, which

reconstructs a graph $A \in \{0,1\}^{n \times n}$ using logit factors $X, Y \in \mathbb{R}^{n \times k}$ via

$$A \approx \sigma(XY^\top). \tag{4}$$

Note that unlike our community-based factorization, the factors of the LPCA model are not nonnegative, and the factorization does not reflect the symmetry of the undirected graph's adjacency matrix. Regardless of the model's interpretability, the following theorem provides a significant guarantee on its expressiveness. We use the following notation: given a matrix $M$, let $H(M)$ denote the matrix resulting from entrywise application of the Heaviside step function to $M$, that is, setting all positive entries to 1, negative entries to 0, and zero entries to $1/2$.

**Theorem 5.1** (Exact LPCA Factorization for Bounded-Degree Graphs ). *Let $A \in \{0,1\}^{n \times n}$ be the adjacency matrix of a graph $G$ with maximum degree $c$. Then there exist matrices $X, Y \in \mathbb{R}^{n \times (2c+1)}$ such that $A = H(XY^\top)$.*

This corresponds to arbitrarily small approximation error in the LPCA model (Equation 4) because, provided such factors $X, Y$ for some graph $A$, we have that $\lim_{s \to \infty} \sigma(sXY^\top) = H(XY^\top) = A$. That is, we can scale the factors larger to reduce the error to an arbitrary extent.

We expand on this result in two ways. First, give a new bound for exact embedding in terms of arboricity, rather than max degree. This increases the applicability to real-world networks, which often are sparse (i.e., low arboricity) and have right-skewed degree distributions (i.e., high max degree). Second, we show that any rank-$k$ LPCA factorization can be converted to our model's symmetric nonnegative factorization with $O(k)$ communities. This extends the guarantees on the LPCA model's power for exact representation of graphs, both the prior guarantee in terms of max degree and our new one in terms of arboricity, to our community-based model as well. In Appendix A.1, we also introduce a natural family of graphs - Community Overlap Threshold (COT) graphs - for which our model's community-based factorization not only exactly represents the graph, but also must capture some latent structure to do so with sufficiently low embedding dimensionality.

**Arboricity bound for exact representation** We will use the following well-known fact: the rank of the entrywise product of two matrices is at most the product of their individual ranks, that is,

$$\text{rank}(X \circ Y) \leq \text{rank}(X) \cdot \text{rank}(Y).$$

**Theorem 5.2** (Exact LPCA Factorization for Bounded-Arboricity Graphs). *Let $A \in \{0,1\}^{n \times n}$ be the adjacency matrix of an undirected graph $G$ with arboricity $\alpha$. Then there exist embeddings $X, Y \in \mathbb{R}^{n \times (4\alpha^2+1)}$ such that $A = H(XY^\top)$.*

*Proof.* Let the undirected graph $A$ have arboricity $\alpha$, i.e., the edges can be partitioned into $\alpha$ forests. We produce a directed graph $B$ from $A$ by orienting the edges in these forests so that each node's edges point towards its children. Now $A = B + B^\top$, and every node in $B$ has in-degree at most $\alpha$.

Let $V \in \mathbb{R}^{n \times 2\alpha}$ be the Vandermonde matrix with $V_{t,j} = t^{j-1}$. For any $c \in \mathbb{R}^{2\alpha}$, $[Vc](t) = \sum_{j=1}^{2\alpha} c(j) \cdot t^{j-1}$, that is, $Vc \in \mathbb{R}^n$ is a degree-$(2\alpha)$ polynomial with coefficients $c$ evaluated at the integers $t \in [n] = \{1, \ldots, n\}$. Let $b_i$ be the $i^{\text{th}}$ column of $B$. We seek to construct a polynomial such that for $t$ with $b_i(t) = 1$, $[Vc_i](t) = 0$, and $[Vc_i](t) < 0$ elsewhere; that is, when inputting an index $t \in [n]$ such that the $t^{\text{th}}$ node is an in-neighbor of the $i^{\text{th}}$ node, we want the polynomial to output 0, and for all other indices in $[n]$, we want it to have a negative output. Letting $N(i)$ denote the in-neighbors of the $i^{\text{th}}$ node, a simple instantiation of such a polynomial in $t$ is $-1 \cdot \prod_{j \in N(i)} (t - j)^2$. Note that since all nodes have in-degree at most $\alpha$, this polynomial's degree is at most $2\alpha$, and hence there exists a coefficient vector $c_i \in \mathbb{R}^{2\alpha}$ encoding this polynomial.

Let $C \in \mathbb{R}^{n \times 2\alpha}$ be the matrix resulting from stacking such coefficient vectors for each of the $n$ nodes. Consider $P = VC \in \mathbb{R}^{n \times n}$: $P_{i,j}$ is 0 if $B_{i,j} = 1$ and negative otherwise. Then $(P \circ P^\top)_{i,j}$ is 0 when either $B_{i,j} = 1$ or $(B^\top)_{i,j} = 1$ and positive otherwise; equivalently, since $A = B + B^\top$, $(P \circ P^\top)_{i,j} = 0$ iff $A_{i,j} = 1$. Take any positive $\epsilon$ less than the smallest positive entry of $P \circ P^\top$. Letting $J$ be an all-ones matrix, define $M = \epsilon J - (P \circ P^\top)$. Note that $M_{i,j} > 0$ if $A = 1$ and $M_{i,j} < 0$ if $A = 0$, that is, $M = H(A)$ as desired. Since $\text{rank}(J) = 1$ and $\text{rank}(P) \leq 2\alpha$, by the bound on the rank of entrywise products of matrices, the rank of $M$ is at most $(2\alpha)^2 + 1$. ∎

**Exact representation with community factorization**   LPCA factors $X, Y \in \mathbb{R}^{n \times k}$ can be processed into nonnegative factors $B \in \mathbb{R}_+^{n \times k_B}$ and $C \in \mathbb{R}_+^{n \times k_C}$ such that $k_B + k_C = 6k$ and

$$BB^\top - CC^\top = \tfrac{1}{2}\left(XY^\top + YX^\top\right). \tag{5}$$

As a rough outline of the argument that follows, we will need $6k$ columns in the new factors $B, C$, up from the $k$ columns in $X, Y$, because accounting for the possible asymmetry in $XY^\top$ will double the required columns, and accounting for the nonnegativity of $B, C$ will triple the required columns. Observe that the left-hand side can only represent symmetric matrices, but $XY^\top$ is not necessarily symmetric even if $H(XY^\top) = A$ for a symmetric $A$. For this reason, we use a symmetrization: let $L = \tfrac{1}{2}\left(XY^\top + YX^\top\right)$. Note that $H(L) = H(XY^\top)$, so if $XY^\top$ constitutes an exact representation of $A$ in that $H(XY^\top) = A$, so too do both expressions for $L$ in Equation 5. Pseudocode for the procedure of constructing $B, C$ given $X, Y$ is given in Algorithm 1. The concept of this algorithm is to first separate the logit matrix $L$ into a sum and difference of rank-1 components via eigendecomposition. Each of these components can be written as $+\mathbf{v}\mathbf{v}^\top$ or $-\mathbf{v}\mathbf{v}^\top$ with $\mathbf{v} \in \mathbb{R}^n$, where the sign depends on the sign of the eigenvalue. Each component is then separated into a sum and difference of three outer products of nonnegative vectors, via the following Lemma 5.3.

**Lemma 5.3.** *Let* $\phi : \mathbb{R} \to \mathbb{R}$ *denote the ReLU function, i.e.,* $\phi(z) = \max\{z, 0\}$. *For any vector* $\mathbf{v}$,

$$\mathbf{v}\mathbf{v}^\top = 2\phi(\mathbf{v})\phi(\mathbf{v})^\top + 2\phi(-\mathbf{v})\phi(-\mathbf{v})^\top - |\mathbf{v}||\mathbf{v}|^\top.$$

*Proof.* Take any $\mathbf{v} \in \mathbb{R}^k$. Then

$$\begin{aligned}
\mathbf{v}\mathbf{v}^\top &= (\phi(\mathbf{v}) - \phi(-\mathbf{v})) \cdot (\phi(\mathbf{v})^\top - \phi(-\mathbf{v})^\top) \\
&= + \phi(\mathbf{v})\phi(\mathbf{v})^\top + \phi(-\mathbf{v})\phi(-\mathbf{v})^\top - \phi(\mathbf{v})\phi(-\mathbf{v})^\top - \phi(-\mathbf{v})\phi(\mathbf{v})^\top \\
&= + 2\phi(\mathbf{v})\phi(\mathbf{v})^\top + 2\phi(-\mathbf{v})\phi(-\mathbf{v})^\top - (\phi(\mathbf{v}) + \phi(-\mathbf{v})) \cdot (\phi(\mathbf{v}) + \phi(-\mathbf{v}))^\top \\
&= + 2\phi(\mathbf{v})\phi(\mathbf{v})^\top + 2\phi(-\mathbf{v})\phi(-\mathbf{v})^\top - |\mathbf{v}||\mathbf{v}|^\top,
\end{aligned}$$

where the first step follows from $\mathbf{v} = \phi(\mathbf{v}) - \phi(-\mathbf{v})$, and the last step from $|\mathbf{v}| = \phi(\mathbf{v}) + \phi(-\mathbf{v})$.  ∎

Algorithm 1 follows from Lemma 5.3 and constitutes a constructive proof of the following theorem:

**Theorem 5.4** (Exact Community Factorization from Exact LPCA Factorization). *Given a symmetric matrix* $A \in \{0, 1\}$ *and* $X, Y \in \mathbb{R}^{n \times k}$ *such that* $A = H(XY^\top)$, *there exist nonnegative matrices* $B \in \mathbb{R}_+^{n \times k_B}$ *and* $C \in \mathbb{R}_+^{n \times k_C}$ *such that* $k_B + k_C = 6k$ *and* $A = H(BB^\top - CC^\top)$.

---

**Algorithm 1** Converting LPCA Factors to Community Factors

---

**Input** logit factors $X, Y \in \mathbb{R}^{n \times k}$
**Output** $B \in \mathbb{R}_+^{n \times k_B}, C \in \mathbb{R}_+^{n \times k_C}$ such that $k_B + k_C = 6k$
      and $BB^\top - CC^\top = \tfrac{1}{2}\left(XY^\top + YX^\top\right)$
1: Set $Q \in \mathbb{R}^{n \times 2k}$ and $\boldsymbol{\lambda} \in \mathbb{R}^{2k}$ by truncated eigendecomposition such that
      $Q \times \text{diag}(\boldsymbol{\lambda}) \times Q^\top = \tfrac{1}{2}(XY^\top + YX^\top)$
2: $B^* \leftarrow Q^+ \times \text{diag}(\sqrt{+\boldsymbol{\lambda}^+})$, where $\boldsymbol{\lambda}^+, Q^+$ are the positive eigenvalues/vectors
3: $C^* \leftarrow Q^- \times \text{diag}(\sqrt{-\boldsymbol{\lambda}^-})$, where $\boldsymbol{\lambda}^-, Q^-$ are the negative eigenvalues/vectors
4: $B \leftarrow \left(\sqrt{2}\phi(B^*); \quad \sqrt{2}\phi(-B^*); \quad |C^*|\right)$   ▷ $\phi$ and $|\cdot|$ are entrywise ReLU and absolute value
5: $C \leftarrow \left(\sqrt{2}\phi(C^*); \quad \sqrt{2}\phi(-C^*); \quad |B^*|\right)$
6: **return** $B, C$

---

As stated in the introduction to this section, Theorem 5.4 extends any upper bound on the exact factorization dimensionality from the LPCA model to our community-based model. That is, up to a constant factor, the bound in terms of max degree from Theorem 5.1 and the bound in terms of arboricity from Theorem 5.2 also apply to our model; for brevity, we state just the latter here.

**Corollary 5.5** (Exact Community Factorization for Bounded-Arboricity Graphs). *Let* $A \in \{0, 1\}^{n \times n}$ *be the adjacency matrix of an undirected graph* $G$ *with arboricity* $\alpha$. *Then there exist nonnegative embeddings* $B \in \mathbb{R}_+^{n \times k_B}$ *and* $C \in \mathbb{R}_+^{n \times k_C}$ *such that* $k_B + k_C = 6(4\alpha^2 + 1)$ *and* $A = H(BB^\top - CC^\top)$.

Note that Corollary 5.5 is purely a statement about the capacity of our model; Theorem 5.2 stems from a constructive proof based on polynomial interpolation, and therefore so too does this corollary. We do not expect this factorization to be informative about the graph's latent structure. In the following Section 6, we will fit the model with an entirely different algorithm for downstream applications.

# 6   Experiments

We now present a training algorithm to fit our model, then evaluate our method on a benchmark of five real-world networks.

## 6.1   Training Algorithm

Given an input graph $\boldsymbol{A} \in \{0, 1\}^{n \times n}$, we find low-rank nonnegative matrices $\boldsymbol{B}$ and $\boldsymbol{C}$ such that the model produces $\tilde{\boldsymbol{A}} = \sigma(\boldsymbol{B}\boldsymbol{B}^\top - \boldsymbol{C}\boldsymbol{C}^\top) \in (0, 1)^{n \times n}$ as in Equation 1 which approximately matches $\boldsymbol{A}$. In particular, we train the model to minimize the sum of binary cross-entropies of the link predictions over all pairs of nodes:

$$R = -\sum\nolimits_{ij} \left( \boldsymbol{A}_{ij} \log(\tilde{\boldsymbol{A}}_{ij}) + (1 - \boldsymbol{A}_{ij}) \log(1 - \tilde{\boldsymbol{A}}_{ij}) \right). \tag{6}$$

We fit the parameters by gradient descent over this loss, as well as $L_2$ regularization of the factors $\boldsymbol{B}$ and $\boldsymbol{C}$, subject to the nonnegativity of $\boldsymbol{B}$ and $\boldsymbol{C}$. This algorithm is fairly straightforward; pseudocode is given in Algorithm 2. This is quite similar to the training algorithm of Chanpuriya et al. (2020), but in contrast to that work, which only targets an exact fit, we explore the expression of graph structure in the factors and their utility in downstream tasks. Regularization of the factors is implemented to this end to avoid overfitting. Though in the main paper we outline and evaluate a non-stochastic version of the training algorithm, it can generalize straightforwardly to a more scalable stochastic version, e.g., by sampling links and non-links for the loss function and using projected SGD. In Appendix A.2, we discuss an industry application of our model to tabular dataset completion, for which we employ such stochastic training. Here, we use the simpler non-stochastic training to isolate the impact of model capacity, which is the focus of this work, as opposed to optimization.

---

**Algorithm 2** Fitting the Constrained Model

---

**Input** adjacency matrix $\boldsymbol{A} \in \{0, 1\}^{n \times n}$, regularization weight $\lambda \geq 0$, # of iterations $I$,
    # of homo/heterophilous communities $k_B/k_C$
**Output** fitted factors $\boldsymbol{B} \in \mathbb{R}_+^{n \times k_B}$, $\boldsymbol{C} \in \mathbb{R}_+^{n \times k_C}$ such that $\sigma(\boldsymbol{B}\boldsymbol{B}^\top - \boldsymbol{C}\boldsymbol{C}^\top) \approx \boldsymbol{A}$
 1: Initialize $\boldsymbol{B}, \boldsymbol{C}$ by setting entries to independent samples of $\mathrm{Unif}(0, {}^1\!/\!\sqrt{k_B}), \mathrm{Unif}(0, {}^1\!/\!\sqrt{k_C})$
 2: **for** $i \leftarrow 1$ to $I$ **do**
 3:   $\tilde{\boldsymbol{A}} \leftarrow \sigma(\boldsymbol{B}\boldsymbol{B}^\top - \boldsymbol{C}\boldsymbol{C}^\top)$
 4:   $R \leftarrow -\sum_{ij} \left( \boldsymbol{A}_{ij} \log(\tilde{\boldsymbol{A}}_{ij}) + (1 - \boldsymbol{A}_{ij}) \log(1 - \tilde{\boldsymbol{A}}_{ij}) \right)$
 5:   $R \leftarrow R + \lambda \left( \|\boldsymbol{B}\|_F^2 + \|\boldsymbol{C}\|_F^2 \right)$
 6:   Calculate $\partial_{\boldsymbol{B},\boldsymbol{C}} R$, the gradient of $R$ w.r.t. $\boldsymbol{B}, \boldsymbol{C}$, via differentiation through Steps 2 to 4
 7:   Update $\boldsymbol{B}, \boldsymbol{C}$ to minimize $R$ using $\partial_{\boldsymbol{B},\boldsymbol{C}} R$, subject to $\boldsymbol{B}, \boldsymbol{C} \geq 0$
 8: **end for**
 9: **return** $\boldsymbol{B}, \boldsymbol{C}$

---

**Implementation details** Our implementation uses PyTorch (Paszke et al., 2019) for automatic differentiation and minimizes loss using the SciPy (Jones et al., 2001) implementation of the L-BFGS (Liu & Nocedal, 1989; Zhu et al., 1997) algorithm with default hyperparameters and up to a max of 200 iterations of optimization. We set regularization weight $\lambda = 10$ as in Yang & Leskovec (2013). We include code in the form of a Jupyter notebook (Pérez & Granger, 2007) demo.

## 6.2   Datasets

We use five fairly common mid-size datasets ranging from around 1K to 10K nodes. The selection of these five datasets is partly based on the presence of ground-truth multi-labels, which allows for evaluating the overlapping clustering methods. Statistics for these datasets, including the degeneracy

Table 1: Statistics of datasets used in our experiments. As in Sun et al. (2019), for YOUTUBE and AMAZON, we take only nodes that participate in at least one of the largest 5 ground-truth communities.

| Name | Reference | Nodes | Edges | Labels | Max Degree | Degeneracy |
|------|-----------|-------|-------|--------|------------|------------|
| BLOG | Tang & Liu (2009) | 10,312 | 333,983 | 39 | 3,992 | 114 |
| YOUTUBE | Yang & Leskovec (2015) | 5,346 | 24,121 | 5 | 628 | 19 |
| POS | Qiu et al. (2018) | 4,777 | 92,406 | 40 | 3,644 | 49 |
| PPI | Breitkreutz et al. (2007) | 3,852 | 76,546 | 50 | 593 | 29 |
| AMAZON | Yang & Leskovec (2015) | 794 | 2,109 | 5 | 29 | 6 |

of each network, are given in Table 1. Note that degeneracy is an upper bound on arboricity. We note that the mid-sized networks used in our empirical work actually underemphasize the significance of our theoretical arboricity bound: see, e.g., the real-world networks in Pashanasangi & Seshadri (2021), which have up to tens of millions of nodes but still have degeneracies at most in the hundreds.

**BLOG** is a social network of relationships between online bloggers; the node labels represent interests of the bloggers. Similarly, **YOUTUBE** is a social network of YouTube users, and the labels represent groups that the users joined.

**POS** is a word co-occurrence network: nodes represent words, and there are edges between words which are frequently adjacent in a section of the Wikipedia corpus. Each node label represents the part-of-speech of the word. **PPI** is a subgraph of the protein-protein interaction network for Homo Sapiens. Labels represent biological states. Finally, **AMAZON** is a co-purchasing network: nodes represent products, and there are edges between products which are frequently purchased together. Labels represent categories of products.

While social networks like the former two in this list are generally dominated by homophily (McPherson et al., 2001), the latter three should exhibit significant heterophily. For co-purchasing networks like AMAZON, depending on the product, two of the same kind of product are generally not co-purchased, e.g., Pepsi and Coke, as discussed in Peysakhovich & Bottou (2021). Though less intuitively accessible, there is also prior discussion of disassortativity in word adjacencies (Foster et al., 2010; Zweig, 2016), as well as in PPI networks (Newman, 2002; Hase et al., 2010).

### 6.3 Results

**Expressiveness** First, we investigate the expressiveness of our generative model, that is, the fidelity with which it can reproduce an input network. In Section 1, we used a simple synthetic network to show that our model is more expressive than others due to its ability to represent heterophilous structures in addition to homophilous ones. We now evaluate the expressiveness of our model on real-world networks. As with the synthetic graph, we fix the number of communities or singular vectors, fit the model, then evaluate the reconstruction error. In Figure 4, we compare the results of our model with those of SVD, BIGCLAM (which is discussed in detail in Section 4), and SYMNMF (Kuang et al., 2015). SYMNMF simply factors the adjacency matrix as $A \approx HH^\top$, where $H \in \mathbb{R}_+^{n \times k}$; note that, like SVD, SYMNMF does not necessarily output a matrix whose entries are probabilities (i.e., bounded in $[0, 1]$), and hence it is not a natural graph generative model like ours and BIGCLAM.

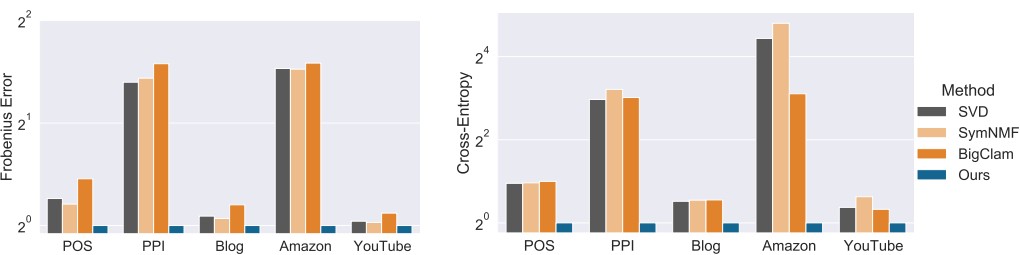

Figure 4: Reconstruction error on real-world networks, relative to our model's error.

For each method, we fix the number of communities or singular vectors at the ground-truth number. For this experiment only, we are not concerned with learning the latent structure of the graph; the only goal is accurate representation of the network with limited parameters. So, for a fair comparison with SVD, we do not regularize the training of the other methods. Our method consistently has the lowest reconstruction error, both in terms of Frobenius error and entrywise cross-entropy (Equation 6). We particularly highlight the improvement over BIGCLAM; the salient difference between these models, both of which are factorization-based, include a nonlinear link function, and assign node communities using nonnegative factors, is the presence of the heterophilous $-CC^\top$ term in our model. Thus, the improvement directly reflects the value of incorporating heterophily into the model. Interestingly, we find the most significant improvement exactly on the three datasets which have been noted to exhibit significant heterophily: POS, PPI, and AMAZON.

**Similarity to ground-truth communities**     To assess the interpretability of clusters generated by our method, we evaluate the similarity of these clusters to ground-truth communities (i.e., class labels), and we compare other methods for overlapping clustering. We additionally compare to another recent but non-generative approach, the VGRAPH method of Sun et al. (2019), which is based on link clustering; the authors found their method to generally achieve state-of-the-art results in this task. For all methods, we set the number of communities to be detected as the number of ground-truth communities. We report F1-Score as computed in Yang & Leskovec (2013). See Figure 5 (left): the performance of our method is competitive with SYMNMF, BIGCLAM, and vGraph.

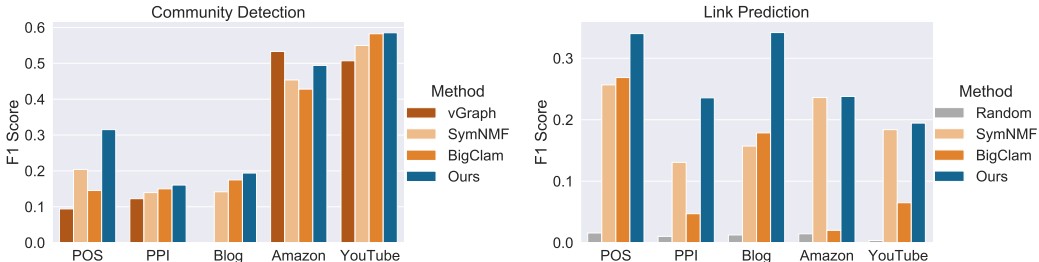

Figure 5: Left: Similarity of recovered communities to ground-truth labels of real-world datasets. (Note: VGRAPH is omitted on BLOG due to memory limitations.) Right: Accuracy of link prediction.

**Interpretable link prediction**     We assess the predictive power of our generative model on the link prediction task. As discussed in Section 3, the link probabilities output by our model are interpretable in terms of a clustering of nodes that it generates; we compare results with our method to those with other models which permit similar interpretation, namely BIGCLAM and SYMNMF. We randomly select 10% of node pairs to hold out, fit the models on the remaining 90%, then use the trained models to predict links between node pairs in the held out 10%. As a baseline, we also show results for randomly predicting link or no link with equal probability. See Figure 5 (right). The performance of our method is competitive with or exceeds those of the other methods in terms of F1 Score.

## 7   Conclusion

We introduce a community-based graph generative model based on symmetric nonnegative matrix factorization which can represent both homophily and heterophily. We add to a prior guarantee of exact representation for bounded degree graphs with a broader guarantee for bounded *arboricity* graphs, and we show that both of these guarantees apply to our more interpretable graph model. We illustrate our model's capabilities with experiments on a synthetic motivating example. Experiments on real-world networks show its effectiveness on several key tasks. Broadly, our results suggest that incorporating heterophily into methods for networks can improve both theoretical grounding and empirical performance, while maintaining interpretability. Future directions include deeper understanding of the expressiveness of low-rank logit models and convergence of training algorithms.

## Acknowledgments and Disclosure of Funding

This project was partially supported by an Adobe Research grant, along with NSF Grants 2046235 and 1763618.

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

# A  Appendix

## A.1  COT Graph Exact Representation

As a theoretical demonstration of the capability of our model to learn latent structure, we additionally show that our model can exactly represent a natural family of graphs, which exhibits both homophily and heterophily, with small $k$ and interpretably. The family of graphs is specified below in Definition 1; roughly speaking, nodes in such graphs share an edge iff they coparticipate in some number of homophilous communities and don't coparticipate in a number of heterophilous communities. For example, the motivating graph described in Section 2 would be an instance of such a graph if an edge occurs between two users iff the two users are from the same city and have different genders.

**Definition 1** (Community Overlap Threshold (COT) Graph). *An unweighted, undirected graph whose edges are determined by an overlapping clustering and a "thresholding" integer $t \in \mathbb{Z}$ as follows: for each vertex $i$, there are two latent binary vectors $\boldsymbol{b}_i \in \{0, 1\}^{k_b}$ and $\boldsymbol{c}_i \in \{0, 1\}^{k_c}$, and there is an edge between vertices $i$ and $j$ iff $\boldsymbol{b}_i \cdot \boldsymbol{b}_j - \boldsymbol{c}_i \cdot \boldsymbol{c}_j \geq t$.*

**Theorem A.1** (Compact Representation of COT Graphs). *Suppose $\boldsymbol{A}$ is the adjacency matrix of a COT graph on $n$ nodes with latent vectors $\boldsymbol{b}_i \in \{0, 1\}^{k_b}$ and $\boldsymbol{c}_i \in \{0, 1\}^{k_c}$ for $i \in \{1, 2, \ldots, n\}$. Let $k = k_b + k_c$. Then, for any $\epsilon > 0$, there exist $\boldsymbol{V} \in [0, 1]^{n \times (k+1)}$ and diagonal $\boldsymbol{W} \in \mathbb{R}^{(k+1) \times (k+1)}$ such that $\left\| \sigma(\boldsymbol{V}\boldsymbol{W}\boldsymbol{V}^\top) - \boldsymbol{A} \right\|_F < \epsilon$.*

*Proof.* Let $t$ be the thresholding integer of the graph, and let the rows of $\boldsymbol{B} \in \{0, 1\}^{n \times k_b}$ and $\boldsymbol{C} \in \{0, 1\}^{n \times k_c}$ contain the vectors $\boldsymbol{b}$ and $\boldsymbol{c}$ of all nodes. Via Equation 2, we can find $\boldsymbol{V}^* \in [0, 1]^{n \times k}$ and diagonal $\boldsymbol{W}^* \in \mathbb{R}^{k \times k}$ such that $\boldsymbol{V}^*\boldsymbol{W}^*\boldsymbol{V}^{*\top} = \boldsymbol{B}\boldsymbol{B}^\top - \boldsymbol{C}\boldsymbol{C}^\top$. Now let

$$\boldsymbol{V} = (\boldsymbol{V}^* \quad \boldsymbol{1}) \qquad \boldsymbol{W} = \begin{pmatrix} \boldsymbol{W}^* & 0 \\ 0 & \frac{1}{2} - t \end{pmatrix}.$$

Then $(\boldsymbol{V}\boldsymbol{W}\boldsymbol{V}^\top)_{ij} = \boldsymbol{b}_i \cdot \boldsymbol{b}_j - \boldsymbol{c}_i \cdot \boldsymbol{c}_j + \frac{1}{2} - t$. Hence $(\boldsymbol{V}\boldsymbol{W}\boldsymbol{V}^\top)_{ij} > 0$ iff $\boldsymbol{b}_i \cdot \boldsymbol{b}_j - \boldsymbol{c}_i \cdot \boldsymbol{c}_j > t - \frac{1}{2}$, which is true iff $\boldsymbol{A}_{ij} = 1$ by the assumption on the graph. Similarly, $(\boldsymbol{V}\boldsymbol{W}\boldsymbol{V}^\top)_{ij} < 0$ iff $\boldsymbol{A}_{ij} = 0$. It follows that

$$\lim_{s \to \infty} \sigma\left(\boldsymbol{V}(s\boldsymbol{W})\boldsymbol{V}^\top\right) = \lim_{s \to \infty} \sigma\left(s\boldsymbol{V}\boldsymbol{W}\boldsymbol{V}^\top\right) = \boldsymbol{A}. \qquad \blacksquare$$

### A.2 Application: Tabular Data Completion

We apply our link prediction algorithm to the task of data completion for categorical tabular data. We first transform such data to a graph as follows: We create a node for each row, as well as a node for each unique category for each of the columns. For each entry of the table, a link occurs between the node for the row and the node for the value in the entry, which is one of the possible categories of the column (e.g., if Bob is American, then a link occurs between the row node for 'Bob' and the category node for 'American'). No links occur between two nodes for rows, or between two nodes for category values. For the negative samples, we select only edges between nodes for rows and nodes for category values; the negative samples would otherwise be dominated by pairs of nodes for rows.

We use two proprietary company datasets datasets which we call COMPANY A and COMPANY B. We use the same experimental setup as described in Section 6.3. Unlike in Section 6.3, we have a restriction on the possible links: each row node must be linked to exactly one of the value nodes for each of the categories. Hence, for each combination of row node and column, we predict a link only with the value node with the highest predicted link probability. We set the number of communities for all methods to 20 and 50 for COMPANY A and COMPANY B, respectively. We employ the stochastic version of our algorithm, based on projected SGD, for which code is also provided. We generally use a batch size of 100; we find that the optimization of BIGCLAM often diverges on COMPANY B with this batch size, so we instead use batches of size 1000 for its optimization. The results are provided in Figure 6. Notably, our proposed approach outperforms comparable (community-based factorization) methods in terms of accuracy. As a baseline, we also show the accuracy when completing the tables by simply selecting the most common categorical value for each column ("PLURALITY"). All methods outperform this baseline on both datasets.

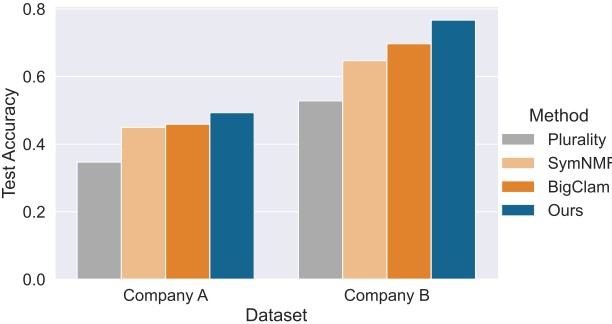

Figure 6: Test accuracy of tabular data completion on two proprietary company datasets (COMPANY A and COMPANY B).

