# OpenReview forum: "Exact Representation of Sparse Networks with Symmetric Nonnegative Embeddings"
_NeurIPS.cc/2023/Conference — NeurIPS 2023 poster_

### Official Review · Reviewer_zLuj · 2023-07-01

**Soundness:** 3 good
**Presentation:** 3 good
**Contribution:** 3 good
**Rating:** 7
**Confidence:** 3

**Summary:**

This paper extends for by Chanpuriya et al (2020) for networks with homophilous and heterophilous edges. They show that their model is able to yield interesting results and they give theoretical underpinnings as well.

**Strengths:**

The model is straightforward but seems to work well. Due do its relative simplicity and its relationship to previous work it is possible to give theoretical guarantees.

**Weaknesses:**

The paper seems to ignore relationships to multilayer and multiplex networks which are also able to capture different types of relationships through different layers. Mason Porter, Peter Mucha and others have written many papers on such networks. There is also a literature on spectral methods for signed networks which can also capture heterophily; Mihai Cucuringu comes to mind. These areas should be mentioned in the related works section.

Some of the presentation is not clear:
In (4) it seems that A is a graph but the right hand side is a probability matrix. How is a 0-1 matrix reconstructed this way?

The overall method is competitive but does not outperform any of the other methods. The conclusion does not give any indication of how the performance could be improved. Is the issue related to independence assumptions? A discussion would be helpful.

**Questions:**

In the motivating example there are 10 cities but the model uses 12 communities. How is the number of communities chosen?

Equation 4: what do the wriggly lines mean?

Theorem 5.1: can it not happen that X Y^T has entries taking the value 0, in which case H would give the value 1/2 and not 0 or 1?

Proof of Theorem 5.2: In my understanding a forest is a collection of trees. What is a partition into forests? It would be good to give a worked example in the appendix. How are the children found? Are the trees supposed to be rooted, choosing a root at random, and then orienting the tree from there? The proof seems to be constructive but it is not clear what the resulting matrices X and Y would be; the final step in the proof seems to be missing.

**Limitations:**

There is no mention of societal impact. For example when assessing credit risk by assigning customers to communities of different risk, would extra caution be advised?

---

> ### Author Rebuttal · Authors · 2023-08-09
>
> Thank you for your review and detailed questions, which we address piecewise.
>
> *On connections to multiplex networks and spectral methods for signed networks*
>
> Thank you for bringing up these interesting connections. Indeed, in some sense, signed graphs seem to capture the idea of heterophilous communities more directly than standard graphs. It seems quite possible that one of the spectral methods you allude to, such as SPONGE, could be slightly altered to detect heterophilous communities instead of homophilous ones. Regarding multiplex networks, this is also an interesting area for connections. At a high level, one of our ideas with this paper is considering types of communities beyond the standard, homophilous ones, and this concept arises more readily when studying types of graphs beyond the standard. This includes multilayer/multiplex graphs, which can have many kinds of edges and thus give more freedom in defining a “community.” We will reconsider these topics and the suggested references, and look into integrating them appropriately into our text.
>
> >The overall method is competitive but does not outperform any of the other methods. The conclusion does not give any indication of how the performance could be improved. Is the issue related to independence assumptions? A discussion would be helpful.
>
> Our method does generally outperform the competitors on the three tasks in the main paper, as well as the one in the appendix; on some tasks and datasets, there is a significant outperformance. That being said, there is certainly room to improve on the numbers. Rather than maximizing empirical performance, one of our main goals with this paper is advancing a conceptual idea and putting it on solid theoretical and empirical footing. This core idea is the benefit of adding to nonnegative embedding models a second, heterophilous embedding in addition to the usual homophilous embedding. This idea could be integrated into larger, more complex, and possibly more performant models, such as deep nonnegative matrix factorization or deep clustering models.
>
> >In the motivating example there are 10 cities but the model uses 12 communities. How is the number of communities chosen?
>
> There are 10 cities but also 2 genders, for a total of 12 (overlapping) communities. The goal with the unsupervised task here is, by detecting both the city and gender communities, to retrieve a 12-dimensional vector for each node/person which is very positive for that person’s city and gender, and roughly 0 for the 10 other entries. In Figure 3, it is shown that our method achieves this, whereas SVD and BigClam have some shortcomings.
>
> >In (4) it seems that A is a graph but the right hand side is a probability matrix. How is a 0-1 matrix reconstructed this way?
>
> > Equation 4: what do the wriggly lines mean?
>
> $\mathbf{A} \approx \sigma(\mathbf{X} \mathbf{Y}^\top)$ denotes approximate equality, as opposed to exact equality as in $\mathbf{A} = \sigma(\mathbf{X} \mathbf{Y}^\top)$. Equation 4 is informally expressing that the goal of LPCA is for $\sigma(\mathbf{X} \mathbf{Y}^\top)$ to be a high probability (near $1$) where $\mathbf{A}$ is $1$ and a low probability (near $0$) where $\mathbf{A}$ is $0$.
>
> >Theorem 5.1: can it not happen that X Y^T has entries taking the value 0, in which case H would give the value 1/2 and not 0 or 1?
>
> Indeed, $H(0) = 1/2$, but the theorem is asserting that there exist $\mathbf{X}$ and $\mathbf{Y}$ such that their product is positive where $\mathbf{A}$ is $1$ and negative where $\mathbf{A}$ is $0$; the theorem is asserting the existence of such $\mathbf{X}$,$\mathbf{Y}$ whose product has no zero entries.
>
> >Proof of Theorem 5.2: In my understanding a forest is a collection of trees. What is a partition into forests? It would be good to give a worked example in the appendix. How are the children found? Are the trees supposed to be rooted, choosing a root at random, and then orienting the tree from there?
>
> Indeed, a forest is a union of disjoint trees. Equivalently, it is an acyclic undirected graph. For our purposes, an undirected graph being a forest means that its edges can be made directed such that each node has at most one incoming edge. This is because each node in a forest participates in a single tree and is either that tree’s root or has a single parent, so orienting each edge towards the parent achieves the desired outcome. A partition into $\alpha$ forests is a partition of the graph’s edges into $\alpha$ undirected acyclic graphs. For our purposes, this means the graph’s edges can be oriented such that each node has at most $\alpha$ incoming edges. We have provided a diagram of an example (see Figure 2 in the PDF of rebuttal figures), and we hope it will help clarify this.
>
> >The proof seems to be constructive but it is not clear what the resulting matrices X and Y would be; the final step in the proof seems to be missing.
>
> We show how to construct a matrix $\mathbf{M} \in \mathbb{R}^{n \times n}$ that exactly encodes the graph in the signs of its entries. We also prove that this matrix is low-rank, specifically that $\text{rank}(\mathbf{M}) = O(\alpha^2)$. By definition of rank, this means there exists a factorization $\mathbf{M} = \mathbf{X} \mathbf{Y}^\top$ for some matrices $\mathbf{X},\mathbf{Y} \in \mathbb{R}^{n \times O(\alpha^2)}$. Constructively, this factorization could be retrieved in many ways, e.g., by eigendecomposition of $\mathbf{M}$. We will add a mention of this to clarify the constructive nature of the proof.

---

> > ### Comment · Reviewer_zLuj · 2023-08-11
> > **The main result from Chanpuriya et al (2020)**
> >
> > Thank you for your reply.
> >
> > Regarding (4), it would be better to rephrase as it sounds as if (4) was a way to construct \tilde{A}, which is not the intention I understand. Instead (1) is the definition of \tilde{A} when B and C are given.
> >
> > Looking more into Theorem 5.1, this is not actually the main result from Chanpuriya et al (2020); they prove it for the function \sigma(s) = max( 0, min (1,x)) instead of the Heaviside function H(x). So why does Theorem 5.1 hold?
> >
> > Thank you for expanding on your notion of a forest. What is the difference between a forest and a tree in your notion? Usually forests are collection of trees, and hence a partition into forests is ambiguous. Your example in the pdf seems to indicate that you equate forest and tree.

---

> > > ### Author Response · Authors · 2023-08-11
> > > **Response to follow-up questions**
> > >
> > > Thank you for following up.
> > >
> > > >Regarding (4), it would be better to rephrase as it sounds as if (4) was a way to construct \tilde{A}, which is not the intention I understand. Instead (1) is the definition of \tilde{A} when B and C are given.
> > >
> > > Thank you for the suggestion. We agree that it would improve clarity to modify (4) by replacing $\mathbf{A} \approx \sigma( \mathbf{X} \mathbf{Y}^\top )$ with $\mathbf{A} \approx \tilde{\mathbf{A}} = \sigma( \mathbf{X} \mathbf{Y}^\top )$. We will make this change and modify the surrounding text appropriately.
> > >
> > > >Looking more into Theorem 5.1, this is not actually the main result from Chanpuriya et al (2020); they prove it for the function \sigma(s) = max( 0, min (1,x)) instead of the Heaviside function H(x). So why does Theorem 5.1 hold?
> > >
> > > This is a great, precise question. The proof of the theorem in Chanpuriya et al. (2020) is actually in terms of the Heaviside nonlinearity $H(z)$ (they call it ($s(z)$), and they just note how it implies the theorem in terms of the clipping nonlinearity $\sigma(z)$ (see the second paragraph of the proof). We lift the theorem from their work without the extra step of this implication because $H(z)$ is more suitable for this paper; we will note this clearly in the revision.
> > >
> > > (The preceding answers the question of why Theorem 5.1 holds, but if you are curious, the implication is as follows: suppose you were given an adjacency matrix $\mathbf{A}$ and found $\mathbf{X},\mathbf{Y}$ such that $H( \mathbf{X} \mathbf{Y}^\top ) = \mathbf{A}$. Then you can scale up the factorization by replacing $\mathbf{X}$ with $\mathbf{X}' = c \cdot \mathbf{X}$ for a large enough positive constant $c$ such that $\sigma( \mathbf{X}' \mathbf{Y}^\top ) = H(\mathbf{X} \mathbf{Y}^\top ) = \mathbf{A}$. This is because, for each entry $z$ in $\mathbf{X} \mathbf{Y}^\top$, either 1) $z$ is negative, in which case $c \cdot z$ is still negative and hence $\sigma(c \cdot z)=H(z)=0$; or 2) $z$ is positive, in which case, for a large enough $c$, you have $c \cdot z \geq 1$ and hence $\sigma(c \cdot z)=H(z)=1$.)
> > >
> > > >Thank you for expanding on your notion of a forest. What is the difference between a forest and a tree in your notion? Usually forests are collection of trees, and hence a partition into forests is ambiguous. Your example in the pdf seems to indicate that you equate forest and tree.
> > >
> > > We use the standard notion of a forest, which is a union of *disjoint* trees. The example in the PDF shows a partition of the edges of a graph into two forests - call them $F_ \text{blue}$ and $F_ {\text{orange}}$. In this case, since we wanted a small example, $F_ \text{blue}$ and $F_ \text{orange}$ both happen to be trees. This does not mean that the union of $F_ \text{blue}$ and $F_ \text{orange}$ itself is a forest, because the union is exactly the original graph, which has a cycle and hence cannot be a forest. (Note that if the definition of forest was *any* union of trees, rather than a union of *disjoint* trees, then the union would be a forest. This is disallowed because the two trees are touching and not disjoint.) Perhaps this thought will help clarify: Suppose we took the example in the PDF and added two extra nodes which have an edge between them but are disconnected from the rest of the graph. These two extra nodes will definitely demand an extra tree to cover (which will just comprise the single edge between them), but they will not demand an extra forest, because this extra tree will be disjoint from both $F_ \text{blue}$ and $F_ {\text{orange}}$ and hence can be added to either forest.

---

> > > > ### Comment · Reviewer_zLuj · 2023-08-12
> > > >
> > > > Thank you for your explanation. Trusting that you will amend the paper as you suggested, I am happy to change my score from 6 to 7.

---

> > > > > ### Author Response · Authors · 2023-08-15
> > > > >
> > > > > We are glad the explanation was helpful. Thank you for following up and reconsidering your score.

---

### Official Review · Reviewer_SSbD · 2023-07-04

**Soundness:** 3 good
**Presentation:** 2 fair
**Contribution:** 2 fair
**Rating:** 5
**Confidence:** 3

**Summary:**

This paper proposes and studies a community-based factorization model for exact representation of sparse networks. The authors extend a prior result on exact factorization which is based on logistic principal component analysis (LPCA), and show that an LPCA factorization can be converted to the proposed community-based graph factorization. The new factorization model explicitly captures both homophily and heterophily among nodes in the graph, and hence it is more interpretable. The authors carry out experiments using a different training algorithm to fit the model, and illustrate the advantage of the proposed decomposition method over existing ones for community detection and link prediction tasks.

**Strengths:**

- The paper provides an intuitive community-based graph factorization model that is able to explicitly capture both homophily and heterophily among nodes in the graph. The expressiveness of the new factorization is backed by a reduction from LPCA which is shown to be able to exactly represent any sparse graph with bounded arboricity. Although it may not be something of extremely high theoretical significance, I liked this aspect of technical development. It is an interesting result to have.

**Weaknesses:**

- There is a clear gap between the exact factorization studied up to Section 5 and the actual training algorithm evaluated in Section 6. Although the authors have mentioned the algorithmic gap explicitly at the end of Section 5. In order to well approximate the heavyside function which is used to achieve exact representation by LPCA, the input of the logit function should have entries that are arbitrarily large in magnitude. However, the training algorithm uses regularization to prevent this. The regularization does exactly the opposite of what is needed for exact representation. Therefore, there is a significant gap between theoretical development and empirical verifications.

- Since the paper studies exact representation, there should be at least an experiment (e.g. the synthetic one) which shows that the reconstruction error goes to 0.

- The overall writing and clarity can be improved. There are a few typos. For example, Line 32, principal components analysis -> principal component analysis. I also could not find where in the paper the term arboricity is clearly defined.

**Questions:**

- For the synthetic experiment shown in Figure 2, what happens if you increase the embedding length further? Will SVD beat your method afterwards? Can you make your method has close to 0 error (since it is supposed to be exact)?

- What happens if you don't regularize your training algorithm for the experiments in Section 6, and how would the methods compare in that case?

**Limitations:**

I cannot find where the authors discuss the limitations or potential negative societal impact of their work, but I think it is fine.

---

> ### Author Rebuttal · Authors · 2023-08-09
>
> Thank you for the review. We appreciate the recognition of our conceptual and theoretical contribution. We are thankful for the suggestions on improving clarity – we will incorporate them. We address the criticisms, which mainly concern the empirical piece.
>
> >There is a clear gap between the exact factorization studied up to Section 5 and the actual training algorithm evaluated in Section 6… The regularization does exactly the opposite of what is needed for exact representation.
>
> We discuss this a bit around Line 290, but exact factorization is not the goal of our empirical contribution. As we note, our training algorithm is quite close to that of Chanpuriya et al., and essentially the whole of their empirical work concerns finding these exact embeddings with such a training algorithm. They report the embedding dimensionalities needed for exact factorization of various networks, among other related information, and we have little to add on that front. We instead use the empirical section to complement our theoretical contributions, and highlight the benefit of adding to nonnegative embedding models a second, heterophilous embedding in addition to the usual homophilous embedding. Indeed, regularization does exactly the opposite of what is needed for exact representation, but it is crucial for our method (as well as the comparison methods) to yield good results on tasks like community detection and link prediction. We also note that the proofs in our theoretical section are constructive and hence provide a totally different “fitting” algorithm that is guaranteed to give an exact factorization, though the resulting embeddings are unlikely to be of practical use.
>
> >For the synthetic experiment shown in Figure 2, what happens if you increase the embedding length further? Will SVD beat your method afterwards? Can you make your method has close to 0 error (since it is supposed to be exact)?
>
> It is quite likely that if we increase capacity by increasing the embedding length that SVD will have lower reconstruction error since it is unregularized. At your suggestion, we ran the training for our method without regularization, evaluated embeddings lengths from 20 to 60, and plotted the results in the PDF of rebuttal figures – see Figure 1. Our method has lower error at all embedding lengths. Further, the fit was essentially exact at length 60. SVD will not achieve zero error until the embedding length equals the number of nodes, 1000, since the adjacency matrix is generally full-rank. We do note that, related to our points above, this is of limited practical significance, since our regularized method captures all the “signal” in the data at embedding length 12, after which there is only noise to fit.

---

> > ### Comment · Reviewer_SSbD · 2023-08-17
> >
> > Thank you for the responses and additional experiments. Most of my questions have been addressed, I slightly raised my score. But I still don't like fact that the actual exact factorization may have very limited practical use, as the authors pointed out in the above. It seems that what works in practice is the training algorithm in Section 6. The theory-to-practice gap is still there.

---

> > > ### Author Response · Authors · 2023-08-17
> > >
> > > We thank you for reviewing, following up, and reconsidering your score. We are glad we have addressed the point from your review about our training algorithm's capability for exact factorization.
> > >
> > > Regarding the point from your comment on the usefulness of exact factorization itself, we briefly expand on this:
> > >
> > > One of the core ideas of the paper is the benefit of adding to nonnegative embedding models a second, heterophilous embedding in addition to the usual homophilous embedding. This allows for the representation of heterophilous structures, which models like BigClam and SymNMF (which lack a heterophilous embedding) will struggle with at practical embedding lengths, as is clearly illustrated in Figures 1-3 for the synthetic plot and also evidenced for real-world networks in Figure 4. The point of the exact embedding result in this context (though it may also be of independent theoretical interest) is to show that our model, and more broadly, nonlinear factorizations with *both* kinds of embeddings, are enough to represent *any* kind of network structure. In other words, representing homophilous and heterophilous structure is enough, and there is no need for, e.g., a third embedding. While the exact embedding results (both theoretical and empirical) are in the regime of no/low regularization and tight fitting, they are reflected in the practically regularized regime which tries to capture signal and ignore noise: as we show throughout the experiments, there is benefit to also capturing heterophilous "signal."

---

> ### Comment · Area_Chair_px9n · 2023-08-17
> **Please provide additional feedback**
>
> Hi,
>
> You seem to have the lowest score for this paper. Could you please acknowledge that you have read the rebuttal and let us know if you still have concerns or not. If not, I would encourage you to raise your score.

---

### Official Review · Reviewer_VwbN · 2023-07-05

**Soundness:** 3 good
**Presentation:** 3 good
**Contribution:** 2 fair
**Rating:** 4
**Confidence:** 4

**Summary:**

This paper proposes to use the logistic PCA model to represent. Paper's main contribution is theoretical – improving the embedding dimensionality bounds from maximum degree to the arboricity of a graph. Overall, I believe the contribution is significant, but currently the paper does not highlight why too much – see more in the weaknesses section. The evaluation is severely limited, which might be forgivable for a more theory-oriented paper.

**Strengths:**

* Theoretical guarantees are interesting to (potentially) broader research community.
* The methods offers grounded interpretability of node embeddings in terms of the community structure.
* The paper is written clearly and should be easy to understand to the general audience.

**Weaknesses:**

* While the theoretical results are interesting, they are not well connected to the literature on either:
 ** Spanning trees and effective resistances and graph sparsification would be exciting to sparsification folks.
 ** Curvature (through bounds on Ollivier-Ricci curvature from effective resistance) would yield interesting connections to GNNs
* The proposed algorithm is not really practical computationally speaking and does not scale to large graphs.
* Experiments are not really indicative of real-world performance of the algorithm and rather focus on provide some free-form support for the theoretical claims of the paper.

**Questions:**

* Would node classification experiments on the datasets like POS/PPI/Blog (these are already present in the experimental section) strengthen the paper? Suggested comparison is some early neural embedding such as DeepWalk. This would open up interest from the subfield of more heuristic node embedding methods.

* Can you detail the comparison with Chanpuriya et al., especially in terms of the proof?

**Limitations:**

Yes; although I would stress limited practicality of the proposed algorithm due to its time complexity.

---

> ### Author Rebuttal · Authors · 2023-08-09
>
> Thank you for the thoughtful review and suggested directions for improvement. We address your criticisms and suggestions as follows.
>
> *On connections of our theoretical results to effective resistance, sparsification, curvature, and more*
>
> Thank you for suggesting these connections. We will look into integrating and discussing them appropriately. We discuss some initial thoughts:
>
> While we focus on exact embeddings in the theoretical results, it would certainly be interesting to look at approximate embeddings, e.g., an embedding that produces a graph which is a spectral sparsifier or spanner of the input graph. Typically, spectral sparsifiers are produced by randomly sampling edges of a graph according to their effective resistances (see e.g. [Spielman, Srivastava 2008]). However, it is also known that a spectral sparsifier can be obtained by sampling $O(\log n / \epsilon^2)$ random spanning trees (see e.g., [Fung, Harvey 2010]). This result ensures that any graph has a spectral sparsifier with arboricity $O(\log n / \epsilon^2)$, and thus can be approximately embedded according to our results with dimension $O(\log^2 n / \epsilon^4)$. Observe that such a result was not possible with just the max degree bound of Chanpuriya et al. as there exist simple examples of graphs where all spectral sparsifiers must have high max degree (e.g., a star graph). It would be interesting to understand if the above approximate embedding bound could be improved via a more refined argument, or extended, e.g. to spanner constructions, many of which should also have low arboricity. It would also be interesting to find practically effective algorithms for finding such approximate embeddings.
>
> Relating to spanners, while our exact embedding does not directly constitute a spanner, it can be thought of as an exact distance oracle, which is a general data structure for approximating distances on graphs (see e.g., [Thorup, Zwick 2005]). However, while our embedding is very compressed, it is not clear if it is useful in this context, since there is no obvious way to *efficiently* calculate distances in the graph given just the embedding, faster than simply reconstructing the full graph. However this could be an interesting direction to pursue in future work.
>
> We are less familiar with Olivier-Ricci curvature, but this also sounds like an interesting connection. Certainly, looking at, e.g., [Sia, Jonckheere, Bogdan 2019], curvature appears to be a powerful tool for detection of traditional, homophilous communities. Possible links to heterophilous community detection and exact embedding are less immediately clear, but are an interesting direction. Please do elaborate if you had some such links in mind.
>
> >The proposed algorithm is not really practical computationally speaking and does not scale to large graphs.
>
> The algorithms we evaluate in the main paper indeed scale quadratically in $n$, and hence are not suitable for large graphs. However, as we note around Line 293, our loss function admits a natural stochastic approximation, so our method transforms straightforwardly to a stochastic version that scales linearly in $n$. We discuss this more in the appendix, where we also evaluate these more scalable algorithms for an industry task. As we state in Line 297, we use the simpler non-stochastic training in the main paper to isolate the impact of model capacity, which is the focus of this work, as opposed to optimization.
>
> >Experiments are not really indicative of real-world performance of the algorithm… Would node classification experiments on the datasets… strengthen the paper? Suggested comparison is some early neural embedding such as DeepWalk.
>
> Indeed, we focus in this work on experiments which complement our theoretical results, which concern non-negative / community-based embeddings. Relatively few methods generate such embeddings, and DeepWalk, node2vec, LINE, etc. are not among them. This is also why we do not evaluate on the node classification task. The concept and theory behind our model is relevant to community detection, which we do evaluate on, and which is an unsupervised analog of node classification.
>
> We do this to keep a tight focus on one of our main goals with this paper, which is advancing a conceptual idea and putting it on solid theoretical and empirical footing. This core idea is the benefit of adding to nonnegative embedding models a second, heterophilous embedding in addition to the usual homophilous embedding. As we note in the response to Reviewer zLuj, this idea could be integrated into larger, more complex, and possibly more performant models, including deep models / GNNs, and including models more suitable for node classification.
>
> >Can you detail the comparison with Chanpuriya et al., especially in terms of the proof?
>
> We start by comparing the proofs of the bounded degree/arboricity results. Both proofs involve polynomial interpolation arguments (this is a common technique for theory about sign rank in general), and we use similar notation where possible for accessibility, which emphasizes similarities. However, they are different in that 1) the central polynomials are different; 2) our proof involves a particular decomposition of $\textbf{A}$ into $\textbf{B}+\textbf{B}^\top$; and 3) the final part our proof involves an entrywise product of polynomial matrices. No analog of (2) or (3) appears in Chanpuriya et al.
>
> More broadly, we think that, while this work certainly builds on Chanpuriya et al., the combination of the arboricity bound above with the non-negative component yields a new conceptual contribution. The result in Chanpuriya et al. is that bounded-degree graphs admit exact low-rank factorization; in this work, we show that for sparse graphs in general (which covers a much wider range of real-world networks), you can exactly express the graph structure in terms of node communities, so long as you also allow for "heterophilous" communities.

---

> > ### Comment · Reviewer_VwbN · 2023-08-14
> >
> > Thanks for the rebuttal. I hope you find connections interesting/useful. In the current form without additional experimental evidence I do not think I should raise the score.

---

> > > ### Author Response · Authors · 2023-08-15
> > >
> > > We thank you for reviewing, responding, and raising interesting connections.
> > >
> > > Related to your comment, we briefly summarize our experiments for the reviewers and chairs:
> > >
> > > - We evaluate NMF models on their ability to represent a synthetic network, as well as several real-world networks. In the rebuttal PDF, we also confirm our model's ability to do exact embedding.
> > >
> > > - We then evaluate on two other *unsupervised learning* tasks (more precisely, tasks without info beyond network structure): 1) community detection and 2) link imputation.
> > >
> > > - In the appendix, to investigate a more realistic application, we evaluate scalable ($O(n)$) versions of the NMF methods for tabular data completion (via link imputation) on industry datasets.
> > >
> > > These experiments span a range in terms of corroborating our theoretical results directly to supporting the graph modeling concepts we introduce. While we believe we have added sufficient experimental support, in order to maintain clarity and readability, we are careful to keep our experimental section closely related to the theoretical and conceptual contributions of the work.

---

### Official Review · Reviewer_gKGt · 2023-07-06

**Soundness:** 3 good
**Presentation:** 4 excellent
**Contribution:** 3 good
**Rating:** 6
**Confidence:** 4

**Summary:**

This paper concerns several main results/themes:

- the authors show a theoretical result on a prior model known as LPCA. Their result indicates that exact factorizations for graphs under the LPCA model is possible under a bounded arboricity assumption, which is more generally applicable than the prior degree-based assumption.

- the authors propose a graph generative model  that uses two non-negative vectors (one for homophily, one for heterophily) per node as well as a non-linear linking function to embed a given graph in an interpretable manner. This models generalizes the prior LPCA model, and has nice properties such as having factorizations that are non-linear, captures heterophily in addition to homophily, and is non-negative.

- they show that the theoretical results that are available for LPCA (including the one that they derived) is applicable to their more general model as well.

- they show promising experimental results on datasets.

**Strengths:**

- Originality: The two original contributions are mainly as follows. 1. the theoretical result on LPCA representations which uses a new arboricity criteria rather than a max degree criteria. 2. The proposed generative model (which is more general than LPCA). To the best of my knowledge, these two contributions have not been priorly proposed. I believe that the level of originality of this article, while not groundbreaking, is sufficient for publication at NeurIPS.

- Quality: The quality of the experiments and the theoretical results are sound and technically solid, to the best of my knowledge. The results of the experiments supports the authors' claims that their proposed approach capture heterophily in an interpretable manner and that in specific cases it outperforms competitors.

- Clarity: the paper is presented in a clear and easy to read manner, which is much appreciated. The authors motivated their results well.

- Significance: Graph embeddings are certainly a relevant area of research nowadays. The authors' contributions have moderate impact, given their expressive, interpretable and more generally applicable model. It is significant enough for publication at NeurIPS.

**Weaknesses:**

- The authors covered virtually all the bases. Overall, I do not see any obvious shortcomings/omissions in the paper. I do have some comments/questions below:

-  From a theoretical and originality perspective, the advances that they make over the prior result (the arboricity vs degree assumption) appears rather marginal (similar to Chanpuriya 2020). Yet a new result is a new result. I do not find this to disqualify this article from being accepted.

- I find a small disconnect between the theoretical results and the proposed methods. The theory says that exact representations exist. The methods provide one way to find a representation. But there seems to be little linking the two: there seems to be no guarantee that what the algorithm actually finds in practice is the exact representation. It is possible that I might be missing something here, and any clarifications from the authors would be appreciated.

**Questions:**

- See above section

**Limitations:**

The main limitations of these classes of methods is scalability and computational complexity. The authors briefly address this in the paper in A.3 for example by mentioning a stochastic training approach. I think the author's discussion of the limitations is sufficient but could be improved, for example, by discussing computational complexity (big Oh run time) in more explicit manners in both the deterministic and the stochastic case.

---

> ### Author Rebuttal · Authors · 2023-08-09
>
> Thank you for the positive review. We address some points you raise below.
>
> >From a theoretical and originality perspective, the advances that they make over the prior result (the arboricity vs degree assumption) appears rather marginal (similar to Chanpuriya 2020).
>
> We believe that, while this work certainly builds on Chanpuriya et al., the combination of the arboricity-based bound with the non-negative component yields a new conceptual contribution. At a high level, the result in Chanpuriya et al. is that bounded-degree graphs admit exact low-rank factorization. In this work, we show that for sparse graphs in general (which covers a much wider range of real-world networks), you can exactly express the graph structure in terms of node communities, so long as you also allow for "heterophilous" communities. Both our theoretical and empirical work ground a core idea that could be integrated into larger and more performant models than the ones we test here, including deep models: the benefit of adding to nonnegative embedding models a second, heterophilous embedding in addition to the usual homophilous embedding. We see Chanpuriya et al. as being more abstracted from practical applications.
>
> >I find a small disconnect between the theoretical results and the proposed methods. The theory says that exact representations exist. The methods provide one way to find a representation. But there seems to be little linking the two: there seems to be no guarantee that what the algorithm actually finds in practice is the exact representation. It is possible that I might be missing something here, and any clarifications from the authors would be appreciated.
>
> Indeed, we provide no theoretical guarantee that our gradient-descent-based training algorithm in the empirical section will find an exact embedding. In some quick experiments in response to Reviewer SSbD (shown in Figure 1 of the rebuttal PDF), we did find that this algorithm can find an exact embedding, at least on the synthetic graph, but we do not focus on this for a few reasons. As we note around Line 290, exact factorization is not the goal of our empirical contribution. Our training algorithm is quite close to that of Chanpuriya et al., and essentially the whole of their empirical work concerns finding these exact embeddings with such a training algorithm. They report the embedding dimensionalities needed for exact factorization of various networks, among other related information, and we have little to add on that front. We instead use the empirical section to complement our theoretical contributions and highlight the core idea we discuss above. We also note that the proofs in our theoretical section are constructive and hence provide a totally different “fitting” algorithm that is guaranteed to give an exact factorization, though the resulting embeddings are unlikely to be of any practical use.
>
> >The main limitations of these classes of methods is scalability and computational complexity. The authors briefly address this in the paper in A.3 for example by mentioning a stochastic training approach. I think the author's discussion of the limitations is sufficient but could be improved, for example, by discussing computational complexity (big Oh run time) in more explicit manners in both the deterministic and the stochastic case.
>
> This is a good point. We note why we focus on the less scalable non-stochastic version in the main paper, and we discuss and evaluate a more scalable stochastic version in the appendix, but we will also specifically note the computational complexities of these versions, which are quadratic and linear in the number of nodes, respectively.

---

> > ### Comment · Reviewer_gKGt · 2023-08-11
> > **reply to authors**
> >
> > I thank the authors for reading and address my comments. I am satisfied with the authors' responses.

---

### Author Rebuttal · Authors · 2023-08-09

We would like to thank the reviewers for their time and helpful suggestions. We address each reviewer with an individual reply. Here, we post the PDF of rebuttal figures that are referenced in these replies.

---

### Comment · Area_Chair_px9n · 2023-08-17
**Comment by the AC**

Dear Authors,

Thank you for your detailed answers and the effort that you put in your rebuttal.

Could the reviewers who haven't responded to the authors please do this as soon as possible.

---

### Decision · Program_Chairs · 2023-09-21

**Decision:**

Accept (poster)

**Comment:**

3/4 reviewers recommend accepting the paper with average score 6. One reviewer rejected it with score 4. I would like to accept this paper as a poster.